# How children and adults keep track of real information when thinking counterfactually

**Jesica Gómez-Sánchez**[ID]*, **José Antonio Ruiz-Ballesteros, Sergio Moreno-Ríos**

Department of Developmental and Educational Psychology, University of Granada, Granada, Spain

* gomezjs@ugr.es

**Data Availability Statement:** The data for both experiments are available at http://sl.ugr.es/dataGS_etal and within the Supporting Information files.

## Abstract

Thinking about counterfactual conditionals such as "if she had not painted the sheet of paper, it would have been blank" requires us to consider what is conjectured (She did not paint and the sheet was blank) and what actually happened (She painted and the sheet was not blank). In two experiments with adults (Study 1) and schoolchildren from 7 to 13 years (Study 2), we tested three potential sources of difficulty with counterfactuals: inferring, distinguishing what is real vs conjectured (epistemic status) and comprehending linguistic conditional expressions ("if" vs "even if"). The results showed that neither adults nor schoolchildren had difficulty in the comprehension of counterfactual expressions such as "even if" with respect to "if then". The ability to infer with both of these develops during school years, with adults showing great ability. However, the third source factor is critical: we found that the key to young children's difficulty with counterfactual thinking was their inability to differentiate real and conjectured information, while adults showed little difficulty with this.

## Introduction

You can easily imagine a mother talking about her son and saying to her young daughter:

i. "If Carlos had worn knee pads to skate, his knees would not have been injured"

But, can the young daughter infer what really happened just from hearing the sentence? Understanding counterfactual conditionals requires her to think of two possibilities [1–3]:

A fact: Carlos did not wear knee pads and his knees were injured

A conjecture: Carlos wore knee pads and his knees were not injured

Moreover, she needs to keep in mind not only that there are two possibilities but also that one of them is real and the other is just a "conjecture". This ability to think about which is the real one and which is the conjectured one is called 'epistemic status' about the possibilities [2]. Here, we study the development of children in tracking that epistemic status and how it contributes to the whole ability to reason counterfactually from subjunctive conditionals.

There are few studies on how adults track the epistemic status (see [4]). In contrast, there has been plenty of research providing evidence that people represent both situations (real and conjectured), with priming (see [1,2]), with comprehension and with inference tasks [4–6]. In

**Funding:** This research was funded by grants from the Spanish Government, Ministry of Economy and Competitiveness (PSI2015-63505-P; PGC2018-095868-B-I00) to SM and the Education, Culture and Sport Ministry (FPU15/05899) to JG. The funders had no role in study design, data collection and analysis, decision to publish, or preparation of the manuscript.

**Competing interests:** The authors have declared that no competing interests exist.

most cases, counterfactual expressions have been formulated using subjunctive conditionals, but not in all (e.g. [7]).

In the text that follows, we review some contradictory results in the literature regarding the development of thinking about "how things could have been" and the experimental strategies used to test this ability. After that, we present how a deductive reasoning framework can help us to study children's difficulty with counterfactuals, not only in making predictions about what is represented but also whether that representation is real or conjectured.

## The development of counterfactual thinking

Until recently, it was unquestioned that primary schoolchildren (from 6 to 12; [8,9]) and even pre-schoolers (from 3 to 5; [10,11]) were able to reason with counterfactual conditionals in the same way and with the same meaning, as adults. However, some studies suggest that their ability could improve during the school years (see [12,13]), and even adults seem to show difficulty in particular situations (e.g. [4,7]).

The discrepancy could be due to the researcher's assumption that if pre-schoolers gave a correct response to a counterfactual conditional, it is because they understand these expressions. However, it is also possible to be correct without understanding the counterfactual conditionals as adults do. Children might understand the counterfactual conditional expression (i) as an indicative conditional, called basic conditional, based on their common knowledge [14]:

ii. "If Carlos wore knee pads to skate, his knees were not injured'"

This was a basic conditional interpretation that would lead to what was called "Basic Conditional Reasoning" [14]. Actually, readers could complete the consequent "not injured", just based on their knowledge about the antecedent: "if Carlos wore knee pads to skate, his knees were . . .?" Researchers found that after presenting children with counterfactual conditionals such as (i), and adding "Carlos did not wear knee pads", children concluded "Carlos's knees were injured" ("modus ponens" inference). Basic conditionals are usually interpreted as biconditionals, and therefore, the logical predictions for basic conditionals are the same as for counterfactual conditionals. As those inferences were consistent with adult counterfactual inferences, researchers thought that young children had the ability to think counterfactually. We think that it is possible that researchers failed to realise that the inferences were produced because children understood the counterfactual expression (i) as the basic conditional (ii), as we mentioned previously. If this is what happened, children's responses to the counterfactual expressions were correct but they may not have understood them in the same way as adults. To be confident about children's counterfactual comprehension, the experimental conditions need to be able to discriminate between counterfactual and basic conditional interpretations.

## Discriminate conditions: Alternative situations and semifactual conditionals

At least two strategies can be used to prevent the interpretation of counterfactuals as just basic conditionals.

One way was provided by Rafetseder, Cristi-Vargas, and Perner [14]. In their "discriminate responses condition", a piece of contextual information was added and a basic conditional interpretation of a counterfactual (ii) led to a wrong answer and, thus, a different one from that obtained using a counterfactual interpretation. For example,

"Yesterday, early in the morning, Carlos was running in the schoolyard and he fell on the ground and hurt his knees. After that, he went skating with his friends".

They asked, "If Carlos had worn knee pads to skate, would his knees have been healthy or injured?" The basic conditional interpretation 'If Carlos wears knee pads to skate, his knees were healthy' would lead to a wrong response as children have to consider that there is another situation that annuls that relation, that is, Carlos's knees were injured even though he wore knee pads to skate because he previously fell on the ground when he was running. Only adolescents and adults could integrate the contextual information to make a Mature Conditional Interpretation that would lead to what Rafetseder, Schwitalla, and Perner [12] called Counterfactual Reasoning.

The second way to create a discriminate response condition for a basic interpretation of counterfactual conditionals is by using semifactual conditional expressions such as:

  iii.  "Even if Carlos had worn knee pads to skate, his knees would have been injured".

As we have seen before, subjunctive conditionals like (i) have generally been proposed by linguists and philosophers (e.g. [15,16]) to envisage the two situations (the conjectured and the real). Semifactual conditionals (iii) are a subtype of counterfactual conditionals, also constructed with the subjunctive mood but usually expressed using concessive terms such as "even if-then", "if-then still", "if-then also" instead of 'if-then' [1,5,6,17]. These expressions appear to cancel an expected causal link between antecedent and consequent: wearing knee pads usually prevents knees being injured but here this causal link is cancelled [18,19]. In this case, the interpretation of semifactuals differs from that of counterfactuals. The real situation in (iii) is that "Carlos did not wear knee pads, and his knees were injured". That is, the antecedent is false (not knee pads) but the consequent remains true (knees injured) instead of false as happens with counterfactuals [20].

Interestingly, a basic conditional interpretation of the conditional (iii) (following a basic conditional reasoning strategy) with the concessive expression "even if Carlos had worn knee pads to skate, his knees would have been. . .?" would lead people to conclude "injured" unlike what would be concluded with "if then" expressions: "not injured". However, people who use a mature interpretation of the counterfactual "if then" in the discriminate conditions proposed by Rafetseder et al. would conclude "injured" in either case because they will obtain a similar representation from the semifactual expression "even if" in (iii) and from the discriminate condition with "if": that is, the compound of the counterfactual expression in (i) and the additional contextual information saying that his knees were already injured.

Using semifactual conditionals, Moreno-Ríos and García-Madruga [21] compared the inferences made by primary schoolchildren from 7 to 11 years old, preadolescents (13–14 years old) and adults. They found that only preadolescents made similar inferences to those of adults. Meanwhile, schoolchildren made inferences as if they understood semifactual conditionals as basic conditionals. Results showed that schoolchildren's ability to make inferences with these conditionals increases until the age of around 14 years.

However, the limitations with counterfactuals of very young children obtained in the previous studies are not uniquely determined by the discriminative structure of the problems. Some interesting studies have shown that children around five years old were able to interpret counterfactual conditionals in the discriminate condition. This was shown by McCormack, Ho, Gribben, O'Connor, and Hoerl [22] and by Nyhout and Ganea [23] (see also [24]). They used two very simple tasks with few objects and the conditionals were about physical causation: the effects on the objects of simple actions. The question was formulated using "counterfactual expressions" in [22] but including the "if. . . still" expression in [24] and [23]. The authors did not mention this fact, but using this expression, the conditional was converted into a semifactual by the use of a concessive. Actually, philosophers of language have debated which is the best semifactual expression, with some championing "even if" [20,25] and others 'if. . .still'

[26]. Moreover, "even if", "if. . .still" and "if . . .also" have the same inferential effects (see Experiment 3, Spanish "aunque/también" in [6]). Therefore, Nyhout and Ganea [23] and Nyhout et al. [24] could be helping children to understand the necessity of the conclusion by using "still" in their formulation, and therefore joining the two strategies for discriminating conditions mentioned above. We cannot know which one of the two strategies was responsible for the results in these experiments. Besides, the grammatical structure of the counterfactual expressions could be playing a role in the difficulty of counterfactual understanding.

Previous studies have tried to disentangle the difficulty of the morphosyntactic expression based on the properties of some languages. Yarbay-Duman, Blom, and Topbas [27] separated two sources of difficulty in counterfactual thinking in Turkish-speaking children: the morphosyntactic complexity of using the subjunctive conditional from the cognitive complexity of considering two different things. Children with some linguistic impairment showed the morphosyntatic complexity effect: they had greater difficulty with counterfactual thinking with subjunctive conditionals than indicative conditionals. In any case, it seems that the use of discriminative conditions is not the only explanation for why schoolchildren have difficulties with some counterfactual problems.

In this study, we used the two strategies previously mentioned (context and "even if") to ensure that people were understanding counterfactuals as such, and not as basic conditionals. We used 'discriminate conditions', such as the one in which Carlos previously fell on the ground when he was running and hurt his knees. To conclude "knees injured", the information provided by the context is needed only in the "if" condition (counterfactuals). It would be unnecessary and reiterative in the "even if" condition (semifactuals) if its grammatical meaning (in any case, the knees were injured) is accessed. Testing adults and children, we can detect whether the grammatical use of "even if" can help them to improve their results, which would imply that they are not actually accessing the grammatical meaning of the counterfactual. If so, we will have a clue that they are using that "morphosyntactic information".

## How people reason with counterfactuals

The study of counterfactual reasoning in children has not been driven by contrasting different theories (see [12,14]). One reason could be that the main theories of conditional reasoning agree about the dual meaning of counterfactuals [28]. Thus, for the mental model theory, each possibility is represented as a mental model: an iconic representation of a situation that captures its basic structure. It can also contain symbolic elements that codify abstract features of the situation, such as negations (¬) or other "mental footnotes" of counterfactual, obligation, belief, etc. (see [29]).

As we have seen before, counterfactual thinking requires us to consider a real situation and another conjectured one [1–3]. It can be induced using counterfactual expressions "if A, then B", such as (i) in which something different to what is said [2]: (conjectured:) 'Carlos wore knee pads and his knees were healthy (not-injured)'; A B actually happened:

(real:) 'Carlos did not wear knee pads and his knees were injured'; ¬A ¬B.

Therefore, the real situation from counterfactuals has a false antecedent and consequent. However, in the real situation of semifactual conditionals (iii), the antecedent is false but the consequent remains true:

(real:) "Carlos did not wear knee pads, and his knees were injured": ¬A B.

As we have seen, children in the study by Rafetseder et al. [12] seemed not to use a counterfactual representation while younger children in McCormack et al. [22] and Nyhout and Ganea [23] did. It is possible that in Rafetseder et al. [12], children only represented the conjectured situation according to the basic conditional reasoning strategy. However, it is also

possible that children in all these studies would be representing both the real and the conjectured possibilities, but in Rafetseder et al.'s study, those representations could be based on a wrong conditional derived from the basic conditional interpretation as they did not consider the contextual information.

In a similar way, the suppositional theory [30,31] maintains that the listeners interpret a pragmatic implication: in the "if" condition (counterfactuals) they understand that the speaker intends to convey the idea of "not-A and not-B" [30], and in the "even if" conditional (semifactuals), the idea that the speaker conveys is "not-A and B" [32,33]. The clearest difference between the mental model theory and the suppositional theory is that only the former establishes that there is a codification of the epistemic status (what is real and what conjectured) as a mental footnote, keeping that information in mind.

A complete understanding of counterfactuals requires distinguishing what is real from what is conjectured and to study this, we need a theory of counterfactuals that makes predictions about how people keep track of the epistemic status. The mental model theory provides us with a clear and useful framework to test the components of counterfactual development. However, there is controversy regarding the epistemic status about what is real and what conjectured. On the one hand, the theory maintains that, due to working memory limitations, people tend to easily forget the mental footnotes, which causes frequent errors in deduction (see [34,35]). On the other hand, it has been proposed that the epistemic labels are not so labile [36,37]. They propose a "permanent label" hypothesis, suggesting that the mental footnotes which allow us to keep track of what is real and what conjectured are not easily forgotten. They support the idea that both situations (the real and the conjectured) are equally understood [38] and accessible [36,37], remaining equally available. Contrary to the "permanent label hypothesis", the mental model theory predicts that people would tend to lose labels of the "epistemic status", which leads to different predictions: inferences which require accessing the content of the models (e.g. the elements "A B") will be more accurate than those that require accessing the real and presupposed labels of those contents.

Therefore, from the mental model theory, two predictions can be derived. First, an improvement with age in the ability to distinguish real and conjectured situations (epistemic status): children's cognitive development is related to an increase in working memory efficiency [39] and therefore fewer mental footnotes will be lost. Also for this reason, the same developmental effect is expected for the epistemic status. Second, if the 'permanent label' [36,37] hypothesis does not hold, only the representations stay and children lose the label indicating what is conjectured and what real, so more errors are expected in detecting the epistemic status than in the inference process: when footnotes are lost, the mental model remains, and therefore although people cannot distinguish what is real and what is conjectured, they can still use the model to make inferences. There is also a specific prediction deriving from how counterfactuals and semifactual conditionals are represented: "the knees were injured" (B) should be more frequently accepted with "Even if" because the fact is present in the two possibilities (conjectured: A B and real: ¬A B), but with "If then", it is only present in one (conjectured: A B; real: ¬A ¬B). However, no differences between the two conditionals are predicted for concluding about the antecedent "Carlos wore knee pads" (A), given that in the representations of both conditionals this element is present just one time.

It is possible that the ability to codify the epistemic status adequately, that is, keeping track of what information is real and what conjectured, could be one of the causes of developmental differences. Ruiz-Ballesteros and Moreno-Ríos [4] showed, in accordance with the standard mental model theory predictions, that even adults, in some demanding tasks in which different conditionals had to be integrated, lost track of what information was real and what conjectured, confusing logical truth (conjectured situation) with empirical truth (real situation) [40].

Lastly, one of the difficulties in comparing children and adults is that frequently, different tasks are used (see discussion in [41]). In the following two experiments, we will try to understand the nature of the difficulty in thinking counterfactually, using the same task. We will test this ability in adults and children. This will allow us to distinguish between basic and counterfactual reasoning, while tracking whether they keep the epistemic label of the information (real and conjectured), as well as whether they can use the linguistic knowledge about subjunctive-concessive expressions ('if' and 'even if').

## Study 1

We test the comprehension of counterfactuals by using stories in which a character utters a sentence with a counterfactual conditional. Only if participants understand it will they be able to answer the final questions correctly. A sample story is: Yesterday, early in the morning, Carlos was running in the schoolyard and he fell on the ground and hurt his knees. Later in the evening, *Carlos's mother saw him skating. Afterwards, she said*: *"If he had worn kneepads to skate, his knees would have been . . ."*

. . . *injured or healthy*? (inferential question).

To test the epistemic question, we say: Remember that *the mother said*: *"if Carlos had worn kneepads to skate. . ." According to this evidence, did the mother see Carlos wearing his kneepads*? *Yes or No*

The story does not say explicitly whether the mother saw the kneepads (just that she saw him skating), but comprehension of the counterfactual expression inform them the fact ("if Carlos had worn his kneepads. . ." which means that actually he was not wearing them).

If participants make a basic conditional interpretation of the counterfactual, they will represent only one mental model (A B). By contrast, if they make a mature counterfactual interpretation, they will represent two mental models with epistemic labels about what is conjectured (A B) and what is real (¬A ¬B) and this includes the contextual information (he previously fell on the ground). Also, the labels that provide us with the information about whether the model is conjectured or presupposed/real can be lost with time. Table 1 shows the representation for

**Table 1. Predictions (correct responses in bold) for the inferential and epistemic questions in Experiments 1 and 2, for the mature (with and without labels) and basic interpretations of counterfactuals with "if" and "even if" counterfactuals based on their initial representations.** See text for details.

| Counterfactual Interpretations | Representations [label] | | | Predictions to Questions: | |
| --- | --- | --- | --- | --- | --- |
| | | | | Epistemic (real) | Inferential |
| *Mature Interpretation* | | | | | |
| If then | *[conject.]* | Kneepad | ¬Injured(*) | | ¬Injured (**Injured***) |
| | *[real]* | ¬Kneepad | Injured | **¬Kneepad** | |
| Even if | *[conject.]* | Kneepad | Injured(*) | | **Injured** |
| | *[real]* | ¬Kneepad | Injured | **¬Kneepad** | |
| *Mature Interpretation (labels lost)* | | | | | |
| If then | | Kneepad | ¬Injured(*) | Kneepad/**¬Kneepad** | ¬Injured (**Injured***) |
| | | ¬Kneepad | Injured | | |
| Even if | | Kneepad | Injured(*) | Kneepad/**¬Kneepad** | **Injured** |
| | | ¬Kneepad | Injured | | |
| *Basic Interpretation* | | | | | |
| If then | | Kneepad | ¬Injured | Kneepad | ¬Injured |
| Even if | | Kneepad | Injured | Kneepad | **Injured** |

(*cancelled by the contextual information = previously injured)

the mature interpretation, with those labels and after losing them, as well as the basic interpretation predicted by the mental model theory (first and second columns).

As was previously shown, children's ability to think counterfactually has to do with 1) the meaning of "counterfactual expressions" but also with 2) the ability to think with two possibilities, one of them false. Our key question here is whether it has also to do with 3) keeping the information about their epistemic status (what is real and what conjectured). In this study, we test adults' sensitivity to these three factors when they think using counterfactual conditionals. Regarding the first, counterfactual conditionals are expressed using subjunctive "If it had happened. . ." expressions and semifactuals with concessive subjunctive "Even if it had happened. . ." conditional expressions. We test whether adults use the information provided by the concessive subjunctive grammatical expressions ("even if") as an aid to their inferred answers. In our example, the initial information creates a discriminate condition by stating a fact:

"Yesterday, early in the morning, Carlos was running in the schoolyard and he fell to the ground and hurt his knees. Later in the evening, Carlos's mother saw him skating."

The following two sentences lead to the same conclusion:

a. "If Carlos had worn knee pads to skate, would his knees have been . . .healthy or injured (correct)?"

b. "Even if Carlos had worn knee pads to skate, would his knees have been . . .healthy or injured (correct)?"

Both expressions lead to the same conclusion: "Carlos's knees were injured", the first because of the contextual information and the second because of that information plus the morphosyntactic aid of "even if". Therefore, only in the "even if" conditional could the inferential question (which will be our first question in the experiment) be correctly solved with the sentence on its own, without the need to consider the fact of the incident in the schoolyard (context; See Inferential Question prediction in Table 1, for Mature I.). Therefore, the improvement of inferences will be a sign that the morphosyntaxis of the subjunctive concessive conditional 'even if' was used to make the inference.

Regarding the second factor, we test whether more correct responses are given with the second expression ("even if"). From the mental model theory, the counterfactual and semifactual conditional expressions lead people to initially represent two facts: one real and another one conjectured. When the right inference is to conclude "Injured", it will be easier with semifactual conditionals, in which the two situations have the same correct conclusion (Injured), than with 'if' counterfactual conditionals, in which only one possibility is consistent with the correct one (Injured). It is therefore with "even if" that the response accuracy should be greater. To avoid a wrong conclusion about the conjectured (Not Injured) with "if then", participants need to consider the contextual information (e.g. the knees were already injured) so as to avoid a Basic conditional interpretation and obtain a Mature conditional interpretation. This is only with "if then"; it is not necessary with "even if" (see Table 1; Mature interpretation, last column on right).

Regarding the third factor, participants have to differentiate between real and conjectured situations. In this case, as we asked for the antecedent, "even if" and "if" counterfactual conditionals have the same two alternative options, based on the mental model theory: Kneepads for the conjectured case and not Kneepads for the real one (see Table 1: Mature I., Representations Column). Therefore, no differences in the difficulty of detecting the real case are expected between the two conditionals.

The new task was based on Rafetseder et al.'s [12] Exp.2 task but adapted to study the epistemic status. As in their study, an inferential question was presented: "If Carlos had worn knee

pads to skate... Would his knees have been healthy or injured?" However, in the present task we also looked for how they distinguish between the inferential and the real state.

There are some differences from the original task. In particular, in half the problems we used "even if" expressions instead of expressions with just "if": "Even if Carlos had worn knee pads to skate...". In addition, in Rafetseder et al.'s task, the causal link between the action (e.g. a girl walked into a room with mud on her shoes) and the result is made explicit (e.g. the floor was dirty), whereas in our task the result can only be inferred from the question itself. Finally, we included a measure of the epistemic status. In order to test it, all the stories included a new character: a police officer who was present and witnessed all the situations described. Thus, participants were asked about what really happened, as in "Did the police officer see Carlos wearing knee pads?"

Predictions about responses for the epistemic and inferential questions are summarised in Table 1, based on the mental model representations. The frequency of responses is expected to be higher when there is only one predicted response. We predicted that:

- Adults would infer according to an appropriate comprehension of the counterfactual conditional (Mature Interpretation rather than a basic conditional one).

- They would identify and differentiate correctly the real and the conjectured situation with some limitations derived by the loss of "mental footnotes" (Table 1, see the two first rows: Mature Int. and Mature Int. labels lost).

- The use of the semifactual conditional "even if" will facilitate the inferential question, referring to the consequent, in comparison with the counterfactual "if" (Table 1, Inference Predictions). However, differences would not be shown in the identification of real and conjectured situations, tested by asking about the antecedent, because they both have the same representation for the antecedent (Table 1, Epistemic Predictions).

## Method

**Participants.** Fifty-four adults between 20 and 30 years old ($M_{age}$ = 24.36; $SD$ = 2.44), thirty-five women ($M_{age}$ = 24.28; $SD$ = 2.41) and nineteen men ($M_{age}$ = 24.57; $SD$ = 2.62), participated. They were all volunteers and spoke Spanish as their first language. The sample was composed of all the students enrolled on a course of developmental psychology who accepted the invitation to participate in the study, choosing this activity from among others, to received course credits. Participants read and filled out a consent form for this study complying with the University Research Ethics Committee guidelines (Comité de ética en investigación humana de la Universidad de Granada specifically approved this study: 178/CEIH/2016). The procedure and the task for this Study, as well as for Study 2 with children, were also approved by the same committee.

**Materials.** Nine stories were created, based on Rafetseder et al. [12] contents to test inferential accuracy and epistemic status in counterfactual reasoning. [In that study, most of the original problems used alternatives to the antecedent but one story implied an alternative to the consequent. We looked for a balance that equalized the numbers of these two kinds of story. The distinction, although interesting, is not included in the paper in order to focus reading on the objectives of the study]. The stories were adapted, controlling factors such as negations and pragmatic implications. Also included in the stories was a character who witnessed the different situations. This allowed us to test the two different epistemic statuses (differentiating conjectured and real situations). Rafetseder and colleagues asked two questions using a different structure: "What would have happened if Susi had taken her shoes off? Would the floor be clean or dirty?" However, we used only one question, as outlined previously, presenting half the stories with "if" counterfactual expressions, and the other half with "even-if" semifactual expressions (see example below).

The workbooks started with a short introduction, which asked participants to take the role of an investigator, using some information provided by a police officer. After this, a practice trial with a story was presented, followed by the 8 experimental stories. Each one consisted of two tasks (questions): the "inferential accuracy" task tested whether participants give correct responses to the counterfactual conditional. The second one, the "epistemic status" task, tested whether they could differentiate real and conjectured situations.

The following is an example of a trial. The manipulation of the conditional can be seen in bold for **even if conditionals** and in brackets (for if conditionals):

*The police officer saw through a window of the room that a child was awake because his alarm clock had just sounded. His sister went into his bedroom to take a toy. Later, the police officer said: "**Even if** (If) his sister had entered silently..."*

Inferential Question

*Would the child have been ... awake (correct) or asleep? (Spanish original version: "¿El niño habría estado...despierto o dormido?")*

Epistemic Status Question

*Remember, the police officer said: "**Even if** (If) his sister had entered silently..." According to this evidence, did the police officer see his sister going in silently? Yes / No (correct) (Spanish original version: "Recuerda que el policía dijo: "**Aunque** (Si) su hermana hubiera entrado en silencio...De acuerdo con esta pista, ¿el policía vio que la hermana entró en silencio? Sí / No")*

The presentation order of the different factors in the eight stories was randomised: the kind of conditional expression (if, even if), the order of the alternative responses, as well as the order of the stories and the correct response to the epistemic status question (yes/no). In half of the cases, the epistemic question was formulated in a complementary way, changing the correct responses between yes and no (e.g. did the police officer see her sister entering noisily? Yes / No). Eight different workbooks with the randomised factors mentioned, were constructed.

**Procedure and design.** The Conditional ('if' vs 'even if') factor was manipulated within-participants. The dependent variable was accuracy in the questions.

The participants were tested individually in a quiet room in a 10 min. session. The experimenter read the stories and the conditional statement aloud and asked the participants first the inferential accuracy question and then the epistemic status question. In the first, they had to complete the conditional with one of the alternatives proposed and in the second, to think about what actually happened, answering yes or no.

## Results

The data for both experiments are available at http://sl.ugr.es/dataGS_etal.

**Inferential response.** We carried out an ANOVA using as independent variable Conditional, with the number of correct responses as dependent variable. As we predicted, the results showed a correct counterfactual understanding with a mean of 89% correct responses. Moreover, as expected, we found a main effect of the Conditional, giving more correct responses with "even if" than with "if" (92% vs 85%; $F(1,53) = 4.93$, $p < .05$, $\eta 2 = .09$), as Table 2 shows.

**Epistemic status.** A second analysis of Conditional was carried out with epistemic status. The epistemic status difficulty (72% correct responses) contrasted with the inferential accuracy previously referred to (89% correct responses) (see Table 3). As predicted, the analysis of conditional between Even if (70%) and if (75%) did not show effects ($F(1,53) = 2.17$, $p = .15$, $\eta 2 = .04$).

However, more correct responses in the epistemic status task were provided by those that responded correctly to the inferential question than by those that did not (90% vs 10%; $\chi 2(1) = 194.92$; $p < .0001$).

**Table 2. Percentage of correct responses in inferential response in Studies 1 with adults and 2 with children.**

| Age Group | Even if | If | M |
|---|---|---|---|
| From 7 to 9 years | 77 (.27) | 61 (.29) | 69 (.28) |
| From 9 to11 years | 86 (.21) | 68(.27) | 77 (.24) |
| From 11 to 13 years | 94 (.11) | 76 (.26) | 85 (.19) |
| M | 85 (.22) | 68 (.28) | |
| Adults | 92 (.13) | 85 (.20) | 89 (.16) |

Means and standard deviations in brackets by Adults, Children's Age groups and Conditional (even if and if).

## Discussion

The results show that, in general, adults present a high ability to conclude correctly from counterfactual conditionals in the inferential question. An interesting and novel result is that adults use the grammatical aid "even if" to give their inferential response. The mental model theory predicts the effect, given that the two mental models in "even if" lead to the correct conclusion ("B"/ Injured in Table 1), while there is only one in the "if then" condition (see Table 1), which needs to consider the contextual information (Mature Counterfactual Interpretation) to establish "B" as the conjectured conclusion. In this way, the study demonstrates the existence of two aid sources in counterfactual reasoning: comprehension of the situation and linguistics ("even if").

Based on the mental model theory, a correct response to the epistemic question requires having a complete representation of counterfactuals that includes the conjectured and the presupposed model as well as their labels. These labels or mental footnotes are easily forgotten [34,42], but when this happens, the mental model remains. It is for this reason that more correct responses are expected in inferences than in detecting epistemic status: the inference can be made by just looking at the elements in the models, but the epistemic question also requires accessing the label of what is true. Accordingly, in this study adults showed some difficulties, with 72% correct responses, this being consistent with other previous results with counterfactuals [4,43]. Also, correct responses to the epistemic inference require discarding the representation derived from what is expressed (the conjectured model) and looking for what is real (the real model) to give a response. It follows that if someone could discard the conjectured model, it is very probable that they could also access it in the inferential question. Thus, most of the correct responses to the epistemic question (what really happened), were preceded by a correct inference based on the conjectured model (90% vs 10%).

## Study 2

In the present study, we test children with the same task as adults. The aim of this study was to evaluate the development of counterfactual reasoning, trying to identify also the factors that

**Table 3. Percentage of correct responses in epistemic status in Studies 1 with adults and 2 with children.**

| Age Group | Even if | If | M |
|---|---|---|---|
| From 7 to 9 years | 39 (.24) | 51 (.25) | 45 (.24) |
| From 9 to 11 years | 39 (.26) | 57 (.23) | 48 (.24) |
| From 11 to 13 years | 57 (.25) | 63 (.26) | 60 (26) |
| M | 45 (.26) | 57 (.25) | |
| Adults | 70 (.24) | 75 (.25) | 72 (.25) |

Means and standard deviations in brackets by Adults, Children's Age groups and Conditional (even if and if).

make deduction in children difficult in the cases detected by Rafetseder et al. [12]. In particular, the study disentangles two potential factors that could be responsible for the development of counterfactual reasoning during the school years: one deductive, the consideration of counterfactual possibilities (inferential responses), and the epistemic status: identifying real and conjectured situations. As far as we know, this last aspect had not been studied previously with children.

Following the mental model theory, counterfactuals and semifactuals, unlike other conditionals, require the construction of multiple mental models, which is cognitively demanding. Children have to create two mental models and inspect models and their elements to make deductions (see [34,35,42]). Also, to identify what is real and what conjectured, they have to keep the mental footnote attached to each model.

Two proposals are contrasted here. On the one hand, the mental model theory establishes that mental footnotes are very easily lost. It was found that the working memory capacities of children from 4 to 15 years old increase linearly [39]. Therefore, due to predictable processing limitations, we would expect an increase with age in both deductive and epistemic detection abilities, but with a particularly poor ability to detect the epistemic status. On the other hand, previous research with adults has supported the hypothesis that the epistemic labels are permanent [36,37]. However, the results in Study 1 question this prediction, at least with that kind of inference task. Therefore, if children can make a mature interpretation, we would expect to fit the predictions of a Mature interpretation without epistemic labels (see Table 1). Of course, it is also possible that some children could interpret counterfactuals just as basic conditionals, and therefore their responses would fit predictions in the last row of Table 1.

Additionally, we aimed to dissociate the ability to think about counterfactual situations from the ability to interpret counterfactual conditional expressions. As shown in Study 1, the mental model theory predicts that the comprehension of "even if A, B" will lead people to draw more correct conclusions of "B" ("Injured" in Table 1) than with "if A, B", based on the way these conditional expressions are represented (see Table 1). Adults seem to use the concessive expression "even if" as assistance to increase the number of correct responses with respect to "if then" subjunctive expressions. This result could show that the linguistic component contributes to good counterfactual reasoning or at least to a good performance in discriminate conditions. The question is whether this happens equally in children.

The mental model theory maintains that the core meaning of conditionals "if A, then B" leads people to obtain an initial representation (A B; [34]). Results in the study of the development of conditionals show that children from the age of seven and adults use the same initial representation (A B; [44]). Therefore, we would expect that children would represent the initial mental model in the same way as adults, and therefore, like adults, they would benefit by using "even if" when they have to conclude "B" from "A". Our hypothesis is that 'even if' will facilitate the inference question of "B" (Injured in Table 1) compared to "if" but that they will not show differences in the epistemic question (about "kneepads" in Table 1), because they have the same representations.

Finally, because the two components of counterfactual reasoning could be related to other cognitive activities or abilities, we took other potentially related measures, such as reading, comprehension and intelligence. It has to be emphasised that linguistic abilities relating to reading ability should not affect the present task, considering that the stories are read by the experimenter. However, children with higher development in these skills, such as good readers, are likely to have more opportunities than poor readers to be exposed to stories with counterfactual scenarios.

## Method

**Participants.** Ninety-six children aged from 7;9 (years; months) to 12;7 ($M_{age}$ = 10;1; $SD$ = 1;3) from a school in Granada participated in this study. There were 49 girls ($M_{age}$ = 10;1; $SD$ = 1;3) and 47 boys ($M_{age}$ = 10;1; $SD$ = 1;4). They were organised into three age-groups: 7–9 years (29 children; Range = 7;9–9;0), 9–11 years (38 children; Range = 9;1–10;9) and 11–13 years (29 children; Range = 11;0–12;7). All participants spoke Spanish as their first language. The sample was composed of all the students from third grade in a school who accepted the invitation to participate in the study. They participated only if their parents gave written consent for this study, complying with the ethical protocol from the University Ethics Committee (Comité de ética en investigación humana de la Universidad de Granada specifically approved this study: 178/CEIH/2016).

**Materials.** Two scales of the Reading Processes Evaluation Battery (PROLEC) [45] were used: the pseudo-words reading (for reading coding) task and the text comprehension task. The first consists of reading different pseudo-words. The second involves two short texts (90 words approximately) and two long texts (130 words approximately). Each had four inferential questions, avoiding answers based only on passive data recall from memory.

Raven's Progressive Matrices [46] is a widely known test of general fluid intelligence. Participants had to select the option which followed the sequence with a total of 60 matrices grouped in five categories (A,B,C,D and E) from the easiest to the most difficult.

Finally, we used the counterfactual reasoning task employed in Study 1, following the same procedure, but in this case, the responses of the children were recorded.

**Procedure and design.** Two factors were manipulated: Conditional ("if" vs "even if") within-participants and Age group (from 7 to 9 years, from 9 to 11 years and from 11 to 13 years) between-participants. Response accuracy in the two inferential and epistemic status question tasks was computed.

The counterfactual reasoning task was carried out in the same conditions as in the first study, that is, individually with an approximate duration of 10 minutes, but in this case in a quiet classroom and recording the responses of the children. In the same session, the PROLEC test was also administered individually, starting with the reading task and followed by the comprehension task. RAVEN was administered in a separate session in groups of four children with a duration of 30 min. approximately. The administration of the tasks was counterbalanced in the following way: half the children did the RAVEN test before the other two tasks and the other half did it afterwards. Also, half the children did the reasoning task before the PROLEC test, and the other half the other way round.

## Results

**Inferential response.** We carried out an ANOVA with Conditional (within-subject variable) and Age (between-subject variable) as the independent variables, and Inferential accuracy as the dependent variable. According to our hypothesis, the results showed more correct responses with "even if" than with "if" in the inferential task (85% vs 68%; F(1,95) = 39.48, p < .001, η2 = .30). Differences were also found with Age group (69% vs 77% vs 85%; F(2,93) = 4.42, p < .05, η2 = .09), revealing a developmental trend in the ability to reason with counterfactual conditionals, as Table 2 shows. The interaction between the two variables was not significant (F(2,93) = .08, p = .92, η2 = .002).

**Epistemic status.** Regarding epistemic status, we carried out a 2 (Conditional) by 3 (Age Group) analysis of variance (ANOVA), using Epistemic status accuracy as dependent variable.

The results showed more correct responses with "if" than with "even if" (57% vs 45%; F (1,93) = 15.41, p < .001, η2 = .14). Moreover, there were differences in Age group (45% vs 48%

vs 60%; $F_{(2,93)}$ = 4.67, p < .01, η2 = .09), showing a developmental lineal trend (polynomial contrast p < .01). The interaction between the two factors was not significant ($F_{(2,93)}$ = 1.30, p = .28, η2 = .03).

However, unlike adults, children do not identify or differentiate correctly between real facts and conjectured facts (see Table 3). The binomial test showed that with "if then", only older children gave responses above chance (63%, binomial test p = .007, given a probability = .5), but not the youngest or the intermediate age group (51% and 57%, binomial test, p = .92 and p = .09 respectively). With "even if" expressions, even older children gave responses by chance (57%, binomial test, p = .16) and the younger children systematically identified what was presented in the conditional as what actually happened (the youngest age group; 39%, binomial test, p = .02; and the intermediate age group 39%, binomial test, p = .01).

Correct responses to the epistemic question were more frequent when correct responses were also given to the inference question for all three age groups (youngest children 62% vs 38%; χ2(1) = 5.5; p < .05; intermediate age-group children 74% vs 26%; χ2(1) = 34.29; p < .0001 and older children 85% vs 15%; χ2(1) = 67.69; p < .0001). This result is to be expected if labels are lost more easily than models, and therefore, children who accessed the label (and responded correctly to the epistemic question) did so because they already had the models (and responded correctly to the inference question).

**Correlation.**   We tested the relationship between the participants' execution in the inferential and epistemic tasks in relation to age (in days), intelligence, comprehension and reading ability measures (note that the stories were read by the researcher and not by the children).

Results show that reasoning abilities correlate with age (Inferential response r = .23; p<0.05; and Epistemic status r = .30; p<0.01) and Intelligence with Inferential response (r = .23; p<0.05). No correlation was found between the two measures of reasoning, consistent with the distinct nature of the two components (r = .09; p>0.05). No other significant correlation was found (see Table 4).

## General discussion

Teachers at school commonly use expressions that require children to understand counterfactual and semifactual conditionals, such as: "if Hitler had won the Second World War, European citizens lives would have been different" and "but even if Hitler had won. . .". The ability to think about what might have happened is essential not only in school but also in everyday life. This is not a simple ability, which may be the reason why there has been a controversy about the age at which children can think counterfactually. Some researchers have found it in

**Table 4.  Correlations between age (A), inferential response (IR), epistemic status (EE), comprehension (C), reading ability (RA) and intelligence (I) in Study 2.**

|     | A      | IR    | EE  | C    | RA  | I   |
|-----|--------|-------|-----|------|-----|-----|
| A   | -      |       |     |      |     |     |
| IR  | .23*   | -     |     |      |     |     |
| EE  | .30**  | .09   | -   |      |     |     |
| C   | -.06   | .20   | .04 | -    |     |     |
| RA  | -.01   | .11   | .01 | -.02 | -   |     |
| I   | .09    | .23*  | .08 | .17  | .06 | -   |

Note

*p< .05

**p< .01

children of 6 years old [22,24], others suggest children of around 3–4 years old possess this ability, as shown with their tasks [10,23], and others maintain that this ability is developed during the school years from 6–12 [12,14]. Even adults seem to show difficulty in particular situations (e.g. [4,7]).

In our research, we did not try to provide a complete account of children's counterfactual reasoning, but just contribute to explaining the difficulties found by Rafetseder et al. in previous studies with schoolchildren when they had to make simple inferences in some very common problems they faced. This was the aim and focus of our research.

To look for possible factors, some studies have used indicative conditionals to create contrary-to-fact situations, such as "if a feather was thrown at a window, then the window was broken" (see e.g. [7,47]). However, counterfactual thinking has been more frequently evaluated using subjunctive conditionals, such as "If A had happened, B would have happened", or using the concessive subjunctive, "Even if A had happened, B would have happened". Development of reasoning with counterfactual conditionals could be influenced by the ability to use and understand these morphosyntactic components [27]. In the present study, we found that this does not seem to be the factor responsible for children's difficulty in reasoning with counterfactuals as, like adults, they used the morphosyntactic information to improve their inferences. They made more correct conclusions when the question was expressed with "even if" semifactuals than with "if then" counterfactuals. Predictions from the mental model theory are consistent with the increase of correct acceptances of the consequent ("B"), because it is present in the two mental models of "even if", but only in one of the "if then" models (see predictions in the inferential question -"Injured"- in Table 1).

The second, and closely related, factor studied here was the inference factor: the ability to think with possibilities (deductive component). Older children, probably with less limited working memory capacity, could draw more correct conclusions than younger children in the "inference question" [48,49], but even the youngest children could make counterfactual inferences. If we concentrate on the 'if' conditionals, we have a similar condition to that used by Rafetseder et al. [12] with very similar and, in fact, slightly better results. Unlike in Rafetseder et al.'s studies, in most of our stories, the real state was not stated explicitly but had to be inferred, and therefore a potential effect of that information on responses, that was present in previous studies, was weakened in the present one. Even so, our findings with the present task suggest, as do Rafetseder et al.'s, that children do not give similar responses to those of adults until the age of 12 (85%, with adults 89%). However, this does not mean that younger children are not able to answer counterfactual questions correctly, with our participants giving 69% accuracy at age 9. Our results are consistent with the proposal of development in multiple stages, as has been proposed previously [7,50]. It is difficult to interpret the different findings in performance in the study of counterfactual reasoning as many features have been proposed as possible causes, such as the use of language or drawings, the executive demands of the task, having a visual support and the kind of cause (physical or not), among others. Here it is important to remember that children of about 5 years could correctly interpret the counterfactual conditional, even in the discriminate condition, when adequate simple and concrete objects were presented in the task [22–24]. Therefore, the difficulty shown by schoolchildren is not due to an inability to think counterfactually per se, but it could be related to their difficulty in following counterfactual alternatives during the comprehension of stories with characters and actions.

The third factor studied is the epistemic component: the ability to codify and access correctly the information about what is real and what is conjectured. The mental model theory maintains that the epistemic status has to be represented as mental footnotes added to one possibility, but these are easily discarded during the inference process (to increase working

memory resources; see [34]). People can then access a particular possibility and make the inferences correctly, but they are more prone to error in finding its mental footnote (which model is the conjectured and which the real one) to draw conclusions about the epistemic status. However, some authors have suggested a 'permanent label' hypothesis based on some previous results [36,37]. They defend the theory that the two labels are not lost: the epistemic status remains. These studies have mainly used comprehension tasks rather than inference tasks and obtained processing measures online [37], with priming tasks [38]. It is possible that in these conditions the epistemic information has not yet been lost. Our results support the standard theory and the ease of forgetting the epistemic status, contrary to what is suggested by the 'permanent label' hypothesis. In accordance with this, adults showed a good ability to think counterfactually about the conjectured situation. However, their ability was not so good when using the epistemic information (mistakes were made in 30% of cases; Study 1). This result is consistent with previous results [4].

A notable result is that schoolchildren showed clear difficulties in this same ability to follow what is presupposed and what conjectured, this being the main source of difficulty in reasoning with counterfactuals. Although this ability improves with development, children's responses were close to random in some conditions. The youngest children scored under 50%, showing that they have serious difficulty in retaining the epistemic mental footnotes. In fact, the youngest group of children and even those under 11 years old, interpreted the semifactual "even if Carlos had worn the knee pads" as if Carlos actually wore knee pads (61% versus 39%). The improvement with age in this ability as well as in the deductive one could be due to the increase in processing resources and working memory capacity [39]: older children can handle more accurately and retain the mental models, their elements, and footnotes but show difficulties as adults because of the footnotes loss.

The differences between counterfactual "if then" and "even if" have also proved interesting. While inferences were better with "even if", as was mentioned and predicted from the mental model theory, no differences were predicted detecting the epistemic status with "if" compared to "even if". Actually, this was the result found in adults, but children made fewer correct responses in "even if" than in "if" problems. One possibility is that the young children do not interpret the linguistic expression "even if" with two models, considering only the conjectured model. However, results with the inferential question suggest that they represent "even if" and "if then" problems in different ways.

One important question that has not been raised here is that Rafetseder et al. [14] used the terms basic conditional reasoning and counterfactual reasoning to distinguish not only the ability to consider contextual information that leads to establish the conclusion, but they also made the assumption that younger children who used basic conditional reasoning did not need to represent two mental models but just one, while only older children and adults who used counterfactual reasoning needed to represent the two models (p. 339). However, there is a possibility that even the young children in Rafetseder et al.'s study (who used a basic conditional reasoning strategy) would have been interpreting the conditional as a counterfactual, but forming a "wrong" conjectured conditional without having considered the contextual information (as was shown in Table 1 in the basic interpretation row). Thus, they would have represented both real and wrongly conjectured possibilities. If this were the case, we would say that even young children can think with two models and this could explain why other studies found no limitation in children of around five years old in counterfactual reasoning tasks. Even so, we think that thinking counterfactually requires construction of the correct conjectured model. Neither Rafetseder et al.'s [12] results (p.401) nor ours can clearly discard the fact that children represent only one model, in fact in the present article we have left this possibility open to readers. However, further research is needed to find out more about this question. In

any case, children's difficulty can be located in their poor ability to distinguish "what is real" from "what is conjectured" which is even lower with "even if" expressions.

Therefore, we could conclude that schoolchildren's difficulty, as found in previous studies with counterfactuals, has at least two potential sources: the deductive component associated with thinking about different possibilities [7,51] that is required to create the real situation ("Carlos did not wear knee pads") negating the conjectured situation ("If Carlos had worn knee pads"), and the epistemic component (what really happened and what was conjectured). Also, we have seen that, for deduction, they do not seem to have great difficulties using and understanding the meaning of the linguistic expressions, and their ability to consider alternatives increases. However, they show a substantial limitation in understanding what is real and what conjectured.

Other cognitive abilities such as literacy and comprehension could be related to the two components of counterfactual thinking studied here. We have seen that intelligence correlates with the inferential component but not with the epistemic knowledge component. Interestingly, the inference and the epistemic components did not correlate, which supports their independence of each other. As it was shown, in other studies pre-schoolers could solve counterfactual tasks with few objects and simple actions. In the present study, schoolers had some difficulties with counterfactuals tested with other tasks. They had to read stories about people and actions, keeping track of the epistemic status of the possibilities. The working memory load and the need to attribute mental representations states in these tasks could have made the counterfactual inferences more difficult. Unfortunately, we did not test other cognitive components such as direct measures of working memory or of theory of mind, which we think could be interesting for future studies.

In short, the focus of children's difficulty with reasoning counterfactually seems to be the epistemic status. The results of this paper suggest that a good candidate factor to improve counterfactual thinking could be directing children's attention to what is real and what conjectured in an explicit way, helping them to keep track of the epistemic status. However, it is unlikely that linguistic expressions used to express the counterfactual are an important factor, at least during school years. More research will help us examine how we could improve this important kind of reasoning.

## Supporting information

**S1 Appendix. Examples of the stories and percentages of correct responses (standard deviations in brackets).** In bold the correct response.
(DOCX)

**S1 Data.**
(XLSX)

## Acknowledgments

We are very grateful to the schools for their assistance with data collection and to the directors, teachers and children for their cooperation. Finally, we wish to thank Ruth Byrne and Cristina Vargas for their helpful comments.

## Author Contributions

**Conceptualization:** José Antonio Ruiz-Ballesteros, Sergio Moreno-Ríos.

**Formal analysis:** Jesica Gómez-Sánchez, Sergio Moreno-Ríos.

**Funding acquisition:** Jesica Gómez-Sánchez, Sergio Moreno-Ríos.

**Investigation:** Jesica Gómez-Sánchez.

**Methodology:** Jesica Gómez-Sánchez, José Antonio Ruiz-Ballesteros, Sergio Moreno-Ríos.

**Project administration:** Sergio Moreno-Ríos.

**Supervision:** Sergio Moreno-Ríos.

**Validation:** Sergio Moreno-Ríos.

**Visualization:** Jesica Gómez-Sánchez, Sergio Moreno-Ríos.

**Writing – original draft:** Jesica Gómez-Sánchez, Sergio Moreno-Ríos.

**Writing – review & editing:** Jesica Gómez-Sánchez, José Antonio Ruiz-Ballesteros, Sergio Moreno-Ríos.

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
