## [Decision Letter · Decision Letter 0]

25 Jun 2020

PONE-D-20-08299

How children and adults keep track of real information when thinking counterfactually

PLOS ONE

Dear Dr. Gómez-Sánchez,

Thank you for submitting your manuscript to PLOS ONE. After careful consideration, we feel that it has merit but does not fully meet PLOS ONE’s publication criteria as it currently stands. Therefore, we invite you to submit a revised version of the manuscript that carefully and systematically addresses all the points raised by the two reviewers during the review process (see below).

We look forward to receiving your revised manuscript.

Kind regards,

Emmanuel Manalo, PhD

Academic Editor

PLOS ONE

Journal Requirements:

2.Thank you for including your ethics statement:  "Comité de ética en investigación humana de la Universidad de Granada (1068/CEIH/2020). Written consent.".   

Please amend your current ethics statement to confirm that your named institutional review board or ethics committee specifically approved this study.

3. We noted in your submission details that a portion of your manuscript may have been presented or published elsewhere. [Partial results of this study could be discussed in a conference during the submission period.]

Additional Editor Comments (if provided):

I have now received detailed comments and suggestions from the two reviewers of your paper. They confirm my initial impression that your research has considerable merit for publication. However, before it can be published, there are numerous aspects of it that require your attention. These are explained clearly in the reviewers' comments and suggestions. I would therefore like to ask that you carefully and systematically work through addressing all of the points they raised before sending a revised version of your paper. Please do not hesitate to contact me if you have any questions at all.

Reviewers' comments:

Reviewer's Responses to Questions

**Comments to the Author**

1. Is the manuscript technically sound, and do the data support the conclusions?

Reviewer #1: Partly

Reviewer #2: Yes

2. Has the statistical analysis been performed appropriately and rigorously? 

Reviewer #1: Yes

Reviewer #2: Yes

3. Have the authors made all data underlying the findings in their manuscript fully available?

Reviewer #1: Yes

Reviewer #2: Yes

4. Is the manuscript presented in an intelligible fashion and written in standard English?

Reviewer #1: Yes

Reviewer #2: No

5. Review Comments to the Author

Reviewer #1: The paper presents two studies using a theoretically well laid-out procedure to explore the need to think about reality when engaging in counterfactual thinking. This issue is an important one: counterfactual thinking is an important topic in its own right and as yet the majority of papers have made more simplistic claims about whether children at a certain age can or cannot think counterfactually. This paper suggests that there may be relatively late important developments. While children and adults comprehended counterfactual expressions such as “even if” in a correct way, especially children struggle to differentiate real and conjectured information. This manuscript is a welcome addition to the counterfactual reasoning literature. The topic is important, as the ability to reason counterfactually is implicated in the acquisition of a range of abilities (e.g., pretense, theory of mind, scientific reasoning, causal inference, and emotional development). The design of the studies is innovative, as the task assesses whether children’s ability to think counterfactually has to do with counterfactual expressions and with their ability to think with two possibilities. Although positively disposed – and I want to stress that I particularly enjoyed the introduction - I have a number of comments, suggestions, and questions.

Line 66. Reference [12] should be exchanged with reference [14] where this label was introduced.

Line 87. I am wondering how the contextual information maps onto the example given in reference [14]. Carlos was running in the schoolyard (which is equivalent to the tall puppet coming to collect the candy from the top shelf) and he injured his knees, because he fell (which is equivalent to the candy ending up in the tall puppet’s room). Later Carlos went skating with his friends, and one can ask “If Carlos had worn knee pads to skate….” – this would mean that a child, who uses BCI, would conclude that his knees would not be injured. Do they mean that the contextual information in reference [14] is the little girl, who could not reach the top shelf (and if she had come the candy would have stayed on the top shelf”? And if so, why is this regarded contextual information? It is not directly linked to the tall boy (not in the same way as it is in the example of Carlos). Could the authors be more explicit about the parallels here, and whether they believe these can be mapped onto each other directly?

Line 107/Table 1. Could the authors be more explicit why the real/presupposed for “If” in the MCI condition is “not injured”?

Line 315: Replace 2 with “

Line 374: I struggled with the clarity of the Epistemic Status Question. It is clear that the police officer saw through the window of the room that the child was awake because the alarm went off. When he claims that “Even if his sister had entered silently…” it is unclear whether the police officer did not see her enter or whether he saw her enter but not silently. I don’t think this matter much, but it causes some insecurity as a participant.

Line 334. It would be good, if the authors could say something about recruitment of participants (also for study 2).

Line 382. The authors say that the order of correct response (yes/no) to the epistemic status question was randomised. As far as I can see the answer to all epistemic status questions is “no” (although I am unsure what the correct answer is for story 8). Perhaps I am missing something here, but I would be grateful if the authors could clarify this to me. In addition, it would be good if the authors could clarify how randomisation was done. Finally, it would help if the authors provided information of which answer was counted correct for each story. I would be glad to see the performance on each story – were there any story effects?

Line 383. Perhaps “workbooks” would be more suitable instead of “questionnaires”.

Line 386. The authors mention that one factor (conditional expression) was manipulated within-subjects. What do they mean with “one” – were the other factors story order, etc.?

Line 403. Table 2 is not needed, since all the results are already fully reported in the text (only the epistemic status for the two conditionals has to be reported in the text).

Line 412. Please report the actual statistics here.

Line 413. Why did the authors not use an ANOVA where inference (even if vs if) and epistemic (even if vs if) were two within subjects-factors? This way they could check for interactions across the two. And it would safe the authors running two independent ANOVAs. The same suggestion applies to study 2.

Line 417. I am surprised the authors call the adults’ results as high ability. I thought it was quite low, particularly in the epistemic status condition. Don’t they worry that the stories may have been unclear to participants (see my comment earlier)?

Line 513. Be more explicit about counterbalancing, and perhaps include these as factors in your ANOVAs.

Line 519. Capitalise “but”

Line 527. Specify which factor was between-subjects and which was within-subjects.

Line 533. Wouldn’t the authors have predicted an interaction, such that there is a developmental trend for “if”, but none for “even if”. If not, I would be grateful to be given an explanation as to why not.

Line 544. How do the authors explain the fact that children found the epistemic status question with “if” easier than with “even if”? This is not what mental model theory would predict.

Line 642. Could the authors be more precise in their conclusion that school-children do not struggle with counterfactual thinking, but with processing complex material. Firstly, why do they conclude this based on their data that show 68% correct answers at the age of 9-11 years, which is very close to chance performance? Secondly, what kind of complex material do children have to deal with, that is not part of counterfactual thinking? The authors themselves say (Line 150) that references [22 – 24] could just be helped by the fact that these studies use expressions like “still” which could have helped children to draw the correct inference.

Line 671. Given that working memory was such a crucial factor for their interpretation, I am wondering why such a task was not included in study 2. What makes the authors reach such a conclusion?

Line 682. But why would children only consider the conjectured model, and if they do, shouldn’t his lead them to the correct answer regardless (since there is no mismatch with the real model)?

Line 694. Wouldn’t the authors agree that in order to think counterfactually, one has to create the correct conjectured possibility? In essence, this means children do not quite know yet, how to build this alternative world, which has to follow certain features. Reference [12] does not claim that children only build one world, they claim that BCR is just not the build according to the constraints.

Reviewer #2: How children and adults keep track of real information when thinking counterfactually

by Gómez-Sánchez, Ruiz-Ballesteros and Moreno-Ríos

Review by S. Khemlani for PLOS ONE

The authors conducted two studies that contrast counterfactual conditionals ("if A had happened, B would have happened") from semifactuals ("even if A had happened, B would have happened"). They spell out the theoretical differences between the two, and posit a hypothesis for specific developmental difficulty in understanding the epistemic status of the clauses in counterfactual and semifactual conditionals. One study on adults shows proficiency in both types of conditional, while a second study on children shows specific difficulty in understanding the epistemic status of counterfactual and semifactual conditionals.

The authors' studies are thorough and well-executed, and their results reveal a fascinating deficiency in the development of counterfactual cognition. Overall, I found the paper to be well-written and concise, and their results and analyses to be thorough. My main concern is with the exposition and introduction of the authors' hypotheses and predictions.

In general, I found myself a bit lost when the authors introduced their predictions (lines 276-308) for Study 1. The authors describe their predictions in the abstract at first. Then they mention their specific manipulations later (e.g., they mention the "inferential question" and the police officer on lines 315-332), but they refer to those manipulations earlier, which caused confusion. I only understood their study later on, after reading lines 356-367. Hence, my main criticism is that the authors should revise the introduction of the study so that it is clear, perhaps by providing examples of the questions they asked, and then justifying why they manipulated what they manipulated.

Likewise, the authors weigh down their exposition with lots of abbreviations: BCR, CFR, BC, MC, MCI, and so on. None of these abbreviations help with clarity, and I suggested removing all of them and opting instead for using the unabbreviated phrases in full.

Finally, I found their Table 1 to be largely unhelpful -- most of the contents of the table were covered in the text itself. What would be more helpful is to provide a table of their specific predictions so that readers, at a glance, can understand what to expect from their studies.

In general, however, all these issues require changes to the exposition. The authors' fundamental results remain strong and convincing, and so I think their paper would make an excellent contribution to the literature.

Minor points

------------

- line 63-66: "This tendency to attribute the complete ability to pre-schoolers was explained because researchers overrode the possibility that children were using an easier strategy that implied understanding the counterfactual conditional (i) as an indicative conditional, called basic conditional, based on their common knowledge [12]" -> This is a run-on sentence, and it was unclear what they authors were trying to convey.

- line 77-79: "They did not realise that the inferences were produced because children understood the counterfactual expression (i) as the basic conditional (ii)." -> This is speculative; the authors write as though it's a factual account, but in fact, it's a post-hoc explanation of what they did. So the authors should qualify it as such.

- line 80-82: "To be confident about children’s counterfactual comprehension conditions need to be able to discriminate between these two types of thinking." -> I didn't follow the authors' argument. I think they mean to say that comprehending a counterfactual conditional may require more than making modus ponens inferences, and that previous studies did not ascertain whether children were capable of reasoning beyond modus ponens.

- line 159: "We cannot know whether one or the two strategies were responsible" -> "We cannot know which one of the two strategies was responsible"

- line 215: "The clearest difference between the mental model theory and the suppositional theory is that only the former establishes that there is a codification of the epistemic status (what is real and what conjectured) as a mental footnote, keeping that information in mind. ... Therefore, the mental model theory provides us with a useful framework to test the components of counterfactual development." -> This is a clear articulation of the difference between the two theories. But I didn't understand how the last sentence was connected to the sentences before it. The authors should explain their reasoning for why it's necessary for a theory of counterfactual development to keep track of epistemic status.

- line 285: "inferential question" -> What do the authors mean by this phrase? They introduce it here without explanation.

- line 301: "so as to avoid a BC interpretation to obtain an MC interpretation" -> What do these abbreviations mean? It wasn't clear.

- line 315: "2Evenif" expressions" -> "2 "even if" expressions"

- lines 411-412: "As predicted, the analysis did not show effects." -> The authors should nevertheless provide the results of the non-significant analysis.

- lines 492-499: The authors have three age groups: < 9, 9-11, and >11. This labeling makes it seem as though 5 year olds were in the <9 age group and 18 year olds were in the >11 age group. Why not simply label the groups as: 7-9, 9-11, 11-13? This provides for a uniform labeling scheme.

6. PLOS authors have the option to publish the peer review history of their article (what does this mean?). If published, this will include your full peer review and any attached files.

Reviewer #1: No

Reviewer #2: **Yes: **Sangeet Khemlani

---

## [Author Response · Author response to Decision Letter 0]

6 Aug 2020

A 'Response to Reviewers' file has been submitted with responses to comments in blue.

Dear Editor,

Thank you for inviting us to submit a revised manuscript.

We want to thank the two reviewers for their comments and for their contributions to improving this paper. We will add below in blue our responses to the reviewers’ comments. 

We have introduced changes in the manuscript following the reviewers’ suggestions and to clarify some points raised by them. Thus, we have exemplified the procedure at the beginning of Study 1. We have changed the text to make predictions of the two studies clearer and we have constructed a new Table 1 for predictions. Also, we have erased Table 2 and integrated the data in the new Tables 2 and 3. In addition, following the reviewers’ recommendations we have included other important clarifications. We have added some data in the supplementary material regarding the frequencies of correct responses for each story. 

We hope that you will consider it suitable for publication in your journal and look forward to hearing from you.

Yours sincerely.

Reviewer #1: The paper presents two studies using a theoretically well laid-out procedure to explore the need to think about reality when engaging in counterfactual thinking. This issue is an important one: counterfactual thinking is an important topic in its own right and as yet the majority of papers have made more simplistic claims about whether children at a certain age can or cannot think counterfactually. This paper suggests that there may be relatively late important developments. While children and adults comprehended counterfactual expressions such as “even if” in a correct way, especially children struggle to differentiate real and conjectured information. This manuscript is a welcome addition to the counterfactual reasoning literature. The topic is important, as the ability to reason counterfactually is implicated in the acquisition of a range of abilities (e.g., pretense, theory of mind, scientific reasoning, causal inference, and emotional development). The design of the studies is innovative, as the task assesses whether children’s ability to think counterfactually has to do with counterfactual expressions and with their ability to think with two possibilities. Although positively disposed – and I want to stress that I particularly enjoyed the introduction - I have a number of comments, suggestions, and questions.

Line 66. Reference [12] should be exchanged with reference [14] where this label was introduced.

Done, sorry.

Line 87. I am wondering how the contextual information maps onto the example given in reference [14]. Carlos was running in the schoolyard (which is equivalent to the tall puppet coming to collect the candy from the top shelf) and he injured his knees, because he fell (which is equivalent to the candy ending up in the tall puppet’s room). Later Carlos went skating with his friends, and one can ask “If Carlos had worn knee pads to skate….” – this would mean that a child, who uses BCI, would conclude that his knees would not be injured. Do they mean that the contextual information in reference [14] is the little girl, who could not reach the top shelf (and if she had come the candy would have stayed on the top shelf”? And if so, why is this regarded contextual information? It is not directly linked to the tall boy (not in the same way as it is in the example of Carlos). Could the authors be more explicit about the parallels here, and whether they believe these can be mapped onto each other directly?

In Rafetseder et al.’s studies (2010, 2013) they used two conditions with two characters. The critical point is that while the tall boy can always reach the sweets, the small girl can only reach the sweets when they are located on the lower shelf, and therefore the sweets end up in her room. The “contextual information” is her size that prevents her reaching the upper shelf and prevents the sweets ending up in her room 

(BCI) There are sweets -> sweets will end up in the girl’s room 

 vs 

(MCI) There are sweets [cannot be reached] -> sweets will not end up in girl’s room 

Children who do not consider this restriction will respond in a different way. In our example, the basic conditional knowledge leads people to understand that with kneepads your knees are safe. However, the contextual restrictions reveal that the knee was already injured because of the previous incident. The “contextual information” prevents his knees being healthy.

(BCI) wearing knee pads -> knees will be healthy vs 

(MCI) wearing knee pads [knees already injured] -> Knees will not be healthy 

Line 107/Table 1. Could the authors be more explicit why the real/presupposed for “If” in the MCI condition is “not injured”?

In the new Table 1, the MCI condition is “injured”. The previous Table 2 was confusing because it did not clearly differentiate the initial representation of the conditional from the contextual representation. We think that the new Table 1 does this more clearly. If participants do not consider the contextual information that prevents the normal course of events, “wearing kneepads” would lead them to conclude (knees are healthy) “not injured”. 

Line 315: Replace 2 with “

Done. Thank you.

Line 374: I struggled with the clarity of the Epistemic Status Question. It is clear that the police officer saw through the window of the room that the child was awake because the alarm went off. When he claims that “Even if his sister had entered silently…” it is unclear whether the police officer did not see her enter or whether he saw her enter but not silently. I don’t think this matter much, but it causes some insecurity as a participant.

We agree with Reviewer 1’s concern about the participants’ “insecurity” about what could have happened. Actually, the fact that adults did not make a perfect execution (it was around 75%) is consistent with Reviewer 1’s observation. The reason we used these ambiguous situations was to make interpretation of the counterfactual expression necessary in order to disambiguate. Participants were not informed about whether the police officer saw it. But they knew that the police officer said: “if the sister had entered silently…”, which means that she actually did not enter silently. Note that the same conclusion does not proceed from a factual interpretation of that expression “if the sister entered silently ….”. In the text, we have mentioned that the story does not explicitly give the information, but participants need to understand the counterfactual expression to know the fact (third paragraph in Study 1).

Line 334. It would be good, if the authors could say something about recruitment of participants (also for study 2).

We have added the information in the Participants sections. We have explained that the adult sample was obtained by an invitation to all the students in a developmental psychology group. In the same way, we invited all the pupils who wanted to participate (and had their parents’ permission) in a school. The only limitation we had was having at least 20 participants per group (following the usual number of participants in other studies). If we had not obtained enough voluntary participants, we would have invited another school or another undergraduate group. 

Line 382. The authors say that the order of correct response (yes/no) to the epistemic status question was randomised. As far as I can see the answer to all epistemic status questions is “no” (although I am unsure what the correct answer is for story 8). Perhaps I am missing something here, but I would be grateful if the authors could clarify this to me. 

This was done by asking in half of the cases for a result (e.g. “going barefoot”) and in the other half for its complementary case (e.g. “wearing shoes”). Regarding the correct response for the epistemic question, it was “yes” in the wording version shown for stories 8 and 3 in the appendix/supporting information. But remember that in half of the cases, the wording was complementary, and therefore the correct response for these stories was “no”. With story 8, in the inference question, the contextual information said that the car door was open because the father forgot to lock it, so, although the mother forgot the car key, the car was open anyway (correct). In the epistemic question, the real (presupposed) fact is the negation of the antecedent, and therefore, when the police officer said that “if/Even if she had forgotten the keys” it indicated that actually she did not forget the keys, and therefore the policeman knew that she had the keys. We have included in bold the correct responses to every story in the S1 Appendix (supporting information). 

In addition, it would be good if the authors could clarify how randomisation was done. Finally, it would help if the authors provided information of which answer was counted correct for each story. I would be glad to see the performance on each story – were there any story effects?

We have included a clarification of the randomisation in the text (lines 424 and 569). We randomised the correct response to the epistemic status question (yes/no) by asking for a result (“did she enter barefoot?”) or for its complement (“did she enter wearing shoes?”; see also previous comment). Also, the frequency of correct responses in each story for adults and children is in the S1 Appendix (supporting information). We computed the frequency of correct responses for each story in every condition for adults and for children. These values are now included in the raw data in the repository (the last tab of the spreadsheet). As you can see, the data shows the same pattern of results for each of the stories: in children, all the average correct responses were higher for “Even if” than for “if” in the inferential questions and the opposite was true for the epistemic questions. We think that this is the most important issue, however as you noticed, there were differences in difficulty between the stories used with children when they had to give the epistemic response. Story 7 was the easiest and Story 2 the hardest. But even with these stories, the same pattern of results is shown as in the rest of the stories. 

Line 383. Perhaps “workbooks” would be more suitable instead of “questionnaires”.

Right. Changed.

Line 386. The authors mention that one factor (conditional expression) was manipulated within-subjects. What do they mean with “one” – were the other factors story order, etc.?

There was only one factor in Study 1. The sentence has been changed. 

Line 403. Table 2 is not needed, since all the results are already fully reported in the text (only the epistemic status for the two conditionals has to be reported in the text).

We have erased Table 2. However, we have included the adult data on a separate line in the new Tables 2 and 3 with the children’s results. Readers can use them as a reference.

Line 412. Please report the actual statistics here.

Done, sorry. 

Line 413. Why did the authors not use an ANOVA where inference (even if vs if) and epistemic (even if vs if) were two within subjects-factors? This way they could check for interactions across the two. And it would safe the authors running two independent ANOVAs. The same suggestion applies to study 2.

We agree that results could be shown more clearly by doing only one ANOVA instead of two. However, we would have some important concerns in doing so: the two measures not only differed in what was asked, they also differed in what they were about: the inference question is about the consequent while the epistemic question is about the antecedent. An interesting and more balanced design would allow the 2x(2x2) design if we asked four questions: one about the antecedents and one about the consequents for both the epistemic and inferential questions. 

Line 417. I am surprised the authors call the adults’ results as high ability. I thought it was quite low, particularly in the epistemic status condition. Don’t they worry that the stories may have been unclear to participants (see my comment earlier)?

You are right, the sentence did not mention that we were referring to the inferential question. We have added that information. The interesting point is that the results are not so good in the epistemic status question, and this is explicitly stated in the next paragraph. Regarding the clarity of the stories, as we mentioned in a previous comment, we think that the reason for not having a higher frequency of correct responses to the epistemic questions is because the stories we included tried to use ambiguous situations that could only be disambiguated from the meaning of the counterfactual expressions (please see previous comment).

Line 513. Be more explicit about counterbalancing, and perhaps include these as factors in your ANOVAs.

In the new test, we describe how the order of administration of the tasks was organised. The number of children per counterbalanced group is too small to show significant differences due to that factor; the analysis does not have enough power to be informative. 

Line 519. Capitalise “but”

done

Line 527. Specify which factor was between-subjects and which was within-subjects.

done

Line 533. Wouldn’t the authors have predicted an interaction, such that there is a developmental trend for “if”, but none for “even if”. If not, I would be grateful to be given an explanation as to why not.

We did not make a clear prediction about the presence of an interaction by age and counterfactual expression (if and even if). Because children’s understanding of counterfactuals increases, we expected a corresponding increase of correct responses in the inferential question in both expressions at the same rate (both are counterfactual expressions). However, if children can grasp the concessive (linguistic clue) meaning of “even if”, an advantage for “even if” counterfactuals would be expected: the concessive could help children accept the conclusion. Actually, that is what happened. We hope the reviewer finds that the new Table 1 makes our predictions clearer.

Line 544. How do the authors explain the fact that children found the epistemic status question with “if” easier than with “even if”? This is not what mental model theory would predict.

Yes, you are right and we think that is an interesting point. Actually, we think that this result throws some doubts on whether young children construct counterfactuals from “even if” in the same way as from “if” (as we said in the discussion). The mental model theory would predict a better result for “even if” in the inferential question (because the right conclusion is in both models: the conjectured and the real one, unlike “if”, which is only in the conjectured one), and therefore, although there is confusion between models, the response will always be correct. Actually, children could be using just one model and they would be right. Regarding the epistemic question, the theory predicts the same antecedent for “if A…” and “even if A”, that is, not-A. In the discussion we explain what we think: 

“…from the mental model theory, no differences were predicted in detecting the epistemic status with “if” compared to “even if”. Actually, this was the result found in adults, but children made fewer correct responses in “even if” than in “if” problems. One possibility is that young children do not interpret the linguistic expression “even if” with two models, considering only the conjectured model. In fact, this is consistent with Rafetseder et al.’s proposal about counterfactual interpretation. However, results with the inferential question suggest that they represent “even if” and “if then” problems in different ways”

Line 642. Could the authors be more precise in their conclusion that school-children do not struggle with counterfactual thinking, but with processing complex material. Firstly, why do they conclude this based on their data that show 68% correct answers at the age of 9-11 years, which is very close to chance performance? Secondly, what kind of complex material do children have to deal with, that is not part of counterfactual thinking? The authors themselves say (Line 150) that references [22 – 24] could just be helped by the fact that these studies use expressions like “still” which could have helped children to draw the correct inference.

You are right and we agree completely with what you say. Maybe we had not explained clearly. What we said was “Likewise, reducing the complexity of the problem by reducing the number of elements could also help them to make more accurate inferences”. We don’t think that children do not struggle with counterfactual thinking, and as you said, the results are clear at this point. That sentence about the use of “complex material” was only speculative and has been erased. It was based on the fact that the articles that obtain good results with young children used a very simple problem (Such as 22 and 24). 

Line 671. Given that working memory was such a crucial factor for their interpretation, I am wondering why such a task was not included in study 2. What makes the authors reach such a conclusion

We think you are right: the study would be more complete if measures of executive function such as inhibitory control and particularly, working memory, and theory of mind were added. A basic result seen in the literature is the increase of WM with age. Moreover, there are some developmental studies on deduction that test the mental model theory about the number of mental models that can be constructed or accessed and do not test working memory, based on previous results of increasing WM by age (e.g. Barrouillet, P., Grosset, N., & Lecas, J. F., 2000; Gauffroy & Barrouillet, 2009; 2011). We restricted the number of tasks in order to guarantee that children had appropriate language development and general intelligence development and assuming that the increase of WM with age was a basic result in research with children and would not be informative (now we do not think so). However, as we agree on the interest of including a direct measure of WM, we have included it in the discussion as something interesting to do in new studies. 

Line 682. But why would children only consider the conjectured model, and if they do, shouldn’t his lead them to the correct answer regardless (since there is no mismatch with the real model)?

We think that with the new Table, what we mean can be seen more easily. If children only represent the conjectured model for “even if” (but the two models for “if then”), the epistemic question is answered with the wrong “Kneepads” while for “if then” they can access the correct “not Kneepads”. Actually, with even if, “kneepads” is the most frequent response for the two youngest groups (61%). 

Line 694. Wouldn’t the authors agree that in order to think counterfactually, one has to create the correct conjectured possibility? In essence, this means children do not quite know yet, how to build this alternative world, which has to follow certain features. Reference [12] does not claim that children only build one world, they claim that BCR is just not the build according to the constraints.

We agree. Thinking counterfactually requires representing the correct conjectured situation (at least). In this paragraph, we have explained our previous idea, questioning whether children represent “two models” vs “one” instead of whether they make correct inferences with counterfactuals, which it seems clear, particularly young, children cannot.

Reviewer #2: How children and adults keep track of real information when thinking counterfactually

by Gómez-Sánchez, Ruiz-Ballesteros and Moreno-Ríos

Review by S. Khemlani for PLOS ONE

The authors conducted two studies that contrast counterfactual conditionals ("if A had happened, B would have happened") from semifactuals ("even if A had happened, B would have happened"). They spell out the theoretical differences between the two, and posit a hypothesis for specific developmental difficulty in understanding the epistemic status of the clauses in counterfactual and semifactual conditionals. One study on adults shows proficiency in both types of conditional, while a second study on children shows specific difficulty in understanding the epistemic status of counterfactual and semifactual conditionals.

The authors' studies are thorough and well-executed, and their results reveal a fascinating deficiency in the development of counterfactual cognition. Overall, I found the paper to be well-written and concise, and their results and analyses to be thorough. My main concern is with the exposition and introduction of the authors' hypotheses and predictions.

In general, I found myself a bit lost when the authors introduced their predictions (lines 276-308) for Study 1. The authors describe their predictions in the abstract at first. Then they mention their specific manipulations later (e.g., they mention the "inferential question" and the police officer on lines 315-332), but they refer to those manipulations earlier, which caused confusion. I only understood their study later on, after reading lines 356-367. Hence, my main criticism is that the authors should revise the introduction of the study so that it is clear, perhaps by providing examples of the questions they asked, and then justifying why they manipulated what they manipulated.

Likewise, the authors weigh down their exposition with lots of abbreviations: BCR, CFR, BC, MC, MCI, and so on. None of these abbreviations help with clarity, and I suggested removing all of them and opting instead for using the unabbreviated phrases in full.

Finally, I found their Table 1 to be largely unhelpful -- most of the contents of the table were covered in the text itself. What would be more helpful is to provide a table of their specific predictions so that readers, at a glance, can understand what to expect from their studies.

In general, however, all these issues require changes to the exposition. The authors' fundamental results remain strong and convincing, and so I think their paper would make an excellent contribution to the literature.

Following your indications, we have changed the exposition in different parts of the article, the introduction, hypothesis and predictions, to make them clearer. We have included an example of the task from the beginning of the introduction of Experiment 1. Also, we have erased the previous Table 1 and constructed a new one with the predictions of the epistemic and inferential questions derived from the different interpretations. All abbreviations have been erased. 

Minor points

------------

- line 63-66: "This tendency to attribute the complete ability to pre-schoolers was explained because researchers overrode the possibility that children were using an easier strategy that implied understanding the counterfactual conditional (i) as an indicative conditional, called basic conditional, based on their common knowledge [12]" -> This is a run-on sentence, and it was unclear what they authors were trying to convey.

Changed. We hope it is now clearer.

- line 77-79: "They did not realise that the inferences were produced because children understood the counterfactual expression (i) as the basic conditional (ii)." -> This is speculative; the authors write as though it's a factual account, but in fact, it's a post-hoc explanation of what they did. So the authors should qualify it as such.

We agree. It has been changed.

- line 80-82: "To be confident about children’s counterfactual comprehension conditions need to be able to discriminate between these two types of thinking." -> I didn't follow the authors' argument. I think they mean to say that comprehending a counterfactual conditional may require more than making modus ponens inferences, and that previous studies did not ascertain whether children were capable of reasoning beyond modus ponens.

The sentence was not clear and has been changed. We have explained that the experimental conditions have to discriminate between children’s basic conditional interpretations and mature counterfactual interpretations.

- line 159: "We cannot know whether one or the two strategies were responsible" -> "We cannot know which one of the two strategies was responsible"

Done. Thank you.

- line 215: "The clearest difference between the mental model theory and the suppositional theory is that only the former establishes that there is a codification of the epistemic status (what is real and what conjectured) as a mental footnote, keeping that information in mind. ... Therefore, the mental model theory provides us with a useful framework to test the components of counterfactual development." -> This is a clear articulation of the difference between the two theories. But I didn't understand how the last sentence was connected to the sentences before it. The authors should explain their reasoning for why it's necessary for a theory of counterfactual development to keep track of epistemic status.

You are right. They were not related. We have changed the text to explain why the mental model theory is a good candidate to guide the test.

- line 285: "inferential question" -> What do the authors mean by this phrase? They introduce it here without explanation.

We have rewritten some parts of the introduction following your recommendations and hope that it is now clear.

- line 301: "so as to avoid a BC interpretation to obtain an MC interpretation" -> What do these abbreviations mean? It wasn't clear.

All abbreviations have been erased and substituted by their complete names. The Basic conditional interpretation and the Mature counterfactual interpretation have been defined earlier. 

- line 315: "2Evenif" expressions" -> "2 "even if" expressions"

Done. Thank you.

- lines 411-412: "As predicted, the analysis did not show effects." -> The authors should nevertheless provide the results of the non-significant analysis.

Done

- lines 492-499: The authors have three age groups: < 9, 9-11, and >11. This labeling makes it seem as though 5 year olds were in the <9 age group and 18 year olds were in the >11 age group. Why not simply label the groups as: 7-9, 9-11, 11-13? This provides for a uniform labeling scheme.

You are right. That is clearer. It has been changed.

---

## [Decision Letter · Decision Letter 1]

14 Oct 2020

PONE-D-20-08299R1

How children and adults keep track of real information when thinking counterfactually

PLOS ONE

Dear Dr. Gómez-Sánchez,

Thank you for submitting your manuscript to PLOS ONE. After careful consideration, we feel that it has merit but does not fully meet PLOS ONE’s publication criteria as it currently stands. Therefore, we invite you to submit a revised version of the manuscript that addresses the minor points raised by Reviewer 1 during the review process (see my comments below and the attached Word document).

We look forward to receiving your revised manuscript.

Kind regards,

Emmanuel Manalo, PhD

Academic Editor

PLOS ONE

Additional Editor Comments (if provided):

Please address the minor points raised by Reviewer 1 in the attached sheet (Review.docx). Please highlight and/or clearly indicate these in your revised manuscript so that I can easily check them (i.e., it may not be necessary to ask Reviewer 1 to check your manuscript again IF I can easily find your corrections). Please indicate also if you make any other changes or if you do something else or nothing in response to some of the suggestions (and provide convincing reasons please).

Reviewers' comments:

Reviewer's Responses to Questions

**Comments to the Author**

1. If the authors have adequately addressed your comments raised in a previous round of review and you feel that this manuscript is now acceptable for publication, you may indicate that here to bypass the “Comments to the Author” section, enter your conflict of interest statement in the “Confidential to Editor” section, and submit your "Accept" recommendation.

Reviewer #1: (No Response)

Reviewer #2: All comments have been addressed

2. Is the manuscript technically sound, and do the data support the conclusions?

Reviewer #1: Yes

Reviewer #2: Yes

3. Has the statistical analysis been performed appropriately and rigorously? 

Reviewer #1: Yes

Reviewer #2: Yes

4. Have the authors made all data underlying the findings in their manuscript fully available?

Reviewer #1: Yes

Reviewer #2: Yes

5. Is the manuscript presented in an intelligible fashion and written in standard English?

Reviewer #1: Yes

Reviewer #2: Yes

6. Review Comments to the Author

Reviewer #1: (No Response)

Reviewer #2: How children and adults keep track of real information when thinking counterfactually

by Gómez-Sánchez, Ruiz-Ballesteros and Moreno-Ríos

Review Revision 1 by S. Khemlani for PLOS ONE

I had reviewed a previous version of this paper. In that review, I noted that the results the authors presented were thorough, theoretically well-motivated, and compelling, and so I recommended acceptance of the paper pending some changes to the exposition for clarity.

In this version, the authors have addressed all of the concerns I had with their previous paper. The resulting manuscript is easier to read, and the tables (particularly Table 1) and naming schemes are much more transparent. They also added some additional explanatory text to lay out their methodology and approach in a clear and concise way. So, my recommendation stands -- this paper is an excellent contribution to the journal.

I found no additional typos or minor edits to make.

7. PLOS authors have the option to publish the peer review history of their article (what does this mean?). If published, this will include your full peer review and any attached files.

Reviewer #1: No

Reviewer #2: **Yes: **Sangeet Khemlani

---

## [Author Response · Author response to Decision Letter 1]

10 Nov 2020

A 'Response to Reviewers' file has been submitted with responses to comments in red.

RESPONSE LETTER

Please read our responses in red below Reviewer 1’s proposals. 

Overall, the manuscript has improved in clarity and readability. I still have some comments and questions, which I would like to see addressed:

Line 19/20. The sentence does not seem grammatically correct

The sentence has been changed. 

Line 27. What is the take home message?

The abstract explains the three sources of difficulty with counterfactuals. The take home message is that it is the epistemic factor and not the inference or the linguistic factor that explains children’s difficulty with counterfactuals. We made this explicit by adding “the third factor is critical: ….”

Line 51. Should this say “in the literature”?

Right. Done. 

Line 83. It is difficult to gage what it means that children “may not have understood them” – my intuition is that they simply process counterfactual questions differently from adults, but not that they do not understand anything. Careful framing is important here.

We agree. We have changed the expression “they may not have understood them,” adding “in the same way as adults”.

Line 317. I struggle to understand the difference between “representations” columns and “predictions to questions” columns. Is the point that adults represent both, while children represent only one situation? If so, I wonder whether this could be visualised in a more accessible way in the Table or elsewhere

That’s right. Adults represent two models, and one question is about the “real” model while the second question is about the “conjectured” model. That is why in the prediction column, each prediction is aligned with the corresponding model in the representation column. We think that Reviewer 1’s difficulty in understanding was due to the spacing lines used. In Table 1, in the Basic interpretation row (children’s representation) there was no line spacing between “if” and “even if”, giving the impression that the representation for a conditional was two models (as Reviewer 1 detected). Now, with the changes in Table 1 in the basic interpretation row, “if” and “even if” have the same line spacing as in the “Mature” rows, and therefore, the one-model representation is visualised in a more accessible way. 

Line 347 ff. Could these bullet-points style be transformed into proper text – it would help readability in my opinion.

Done.

Line 350. I am not sure I understand how the causal link is/isn’t made explicit.

There is a simple difference between Rafetseder et al.’s studies and our study. For example, they explicitly mentioned that a girl went into a room with mud on her shoes (cause), and the floor was dirty (consequence). In our study we do not explicitly inform about the consequence (participants need to infer it). Later, we question what would have happened if the cause had been present (would the consequence have happened?). We have included the example in the text. 

Line 367. It is not quite clear to me based, on the manuscript, why only the representation for the antecedent seems crucial in identifying what is real and what is conjectured (at least, this is how I read this sentence). Is it because only the antecedent was provided in the question?

Yes, it is because the epistemic question in the task asks about the antecedent. The paragraph refers to the task’s differential predictions about how “if then” and “even if” are represented. The prediction is that the real and the conjectured models are the same (unlike what happens with the consequent). 

In that paragraph, we added the text “referring to the consequent” and “tested by asking about the antecedent” to make this clearer.

Line 384. Does the sentence in [] relate to [12] or to the current study?

It is related to [12]. At the beginning of the text, we have added “In that study” to avoid ambiguity.

Line 394/397. The switch from past tense to present tense seems a bit odd here.

Agreed. Changed. 

Line 442/449. I am a bit puzzled that a difference of 7% is significant but a difference of 16% isn’t, despite seemingly similar SDs. Also Table 2 shows that numbers ending .5 are sometimes rounded up (89) while in Table 3 such numbers are rounded down (72).

This is a misunderstanding: the percentages shown here were for the epistemic measure (average 72.5%) and the inferential measure (previously analysed in the previous paragraph; average 88.5%) just to show the great difference between these measures (16%). The analysis referred to in that paragraph as without significant differences is only in the epistemic measure as it is labelled in the epigraph (conditional variable: Even if vs If; that is 70% vs 75%). The text has been clarified in this respect.

Regarding the rounding, it may seem strange, but it is correct. It can be seen in the raw data; the two values are: 0.8866 (in text 88.5% and in table 89%) and .7245 (in text 72.5% and in table 72%). To avoid the same misperception by readers, we present data in the text without decimals as in the table.

Line 448. How can it be justified that responses from “even if” and “if” are averaged, when they have just been shown to be significantly different? Wouldn’t it make sense to make separate predictions for each in comparison to the epistemic status?

Please see previous point. There are no differences between “even if” and “if”, as the epistemic analysis showed. 

Line 475. Why is it probable that – if somebody can discard the conjectured model in the epistemic question – they can access that model in the inferential question?

As mentioned in that section, two basic predictions of the mental model theory are that 1) adults represent counterfactuals with two models, and that 2) the labels of these models (conjectured and real) are easily lost, making it difficult to distinguish between the two models. If a participant correctly detects the “conjectured” model in the first (inferential) question (discarding the wrong epistemic one), it is because the label is accessible. In this case, the prediction of the theory is straightforward: that participants will access the epistemic model in the second question (because the label is accessible). This is more probable than the participant having taken the wrong model in the first question (because the label would not have been accessible). 

*** From this point, the line numbering of the document referred to by Reviewer 1 corresponds with the document “with track changes”. 

Line 691. Not every researcher would agree that indicative conditionals are counterfactual – it is something philosophers would agree with, but developmental findings suggest there is a huge difference in the processing of indicative and subjunctive conditionals.

We agree. There are well documented differences in the processing of indicative and subjunctive conditionals in children and also in adults. But we do not discuss here whether the use of the indicative to create counterfactual situations is appropriate. It is a fact. We just mention that some authors use factual conditionals to create counterfactual situations. The best example is Henry Markovits. We cite two of his studies on that point. To make the reading less “odd”, we have changed “counterfactual” to “contrary-to-fact” at this point. 

Line 714. Rafetseder & Perner (2010) show that reality biases are very rare beyond 4 year of age in counterfactual tasks. I doubt that a reduction in reality bias is the reason why children performed better in the current study. 

The term “reality bias” could be misleading, as Reviewer 1 mentioned, and has been changed. This term has been used (and questioned by some authors) to explain the difficulty of preschoolers in inhibiting knowledge about the reality (the actual outcome) in false belief tasks. Here we refer to the effect of presenting a result explicitly vs requiring participants to infer it. Only the fact of presenting that information has an effect on the probability of recovering that information, influencing their later responses (or judgments). As you know, this has been extensively studied and shown, for example, with the label of hindsight bias, in children and adults. 

Line 728. Could the authors be more precise in what they mean with “difficulty processing complex material”? 

In contrast with tasks used in other previous studies (22-24) with children under 5 that include few objects, this task includes stories with characters and actions that participants need to follow. The sentence has been changed: “the difficulty shown by schoolchildren is not due to an inability to think counterfactually per se, but it could be related to the difficulty in following counterfactual alternatives during the comprehension of stories with characters and actions”

Line 766. What proposal are the authors referring to here?

As we said at the end of the general discussion, “Neither Rafetseder et al.’s [12] results (p.401) nor ours can clearly discard the fact that children represent only one model; in fact in the present article we have left this possibility open to readers”. In the line referred to by the Reviewer, we mentioned a possibility referred to by Rafetseder et al. that the BCR is not the build model. This is consistent with representing just one model. However, that is not explicitly stated by the authors, and after reading a previous concern of Reviewer 1 (previous round), we have erased the sentence. Our argument does not need to attribute the proposal to Rafetseder et al. 

Line 809. How could testing WM and ToM help shed light on the current results? This needs to be made clearer.

We have added the following lines to clarify the interest of the two measures: As shown, in other studies, preschoolers could solve counterfactual tasks with few objects and simple actions. In the present study, schoolers had some difficulties with counterfactuals tested with other tasks. They were given stories about people and actions and needed to keep track of the epistemic status of the possibilities. The working memory load and the need to attribute mental representation states in these tasks could have made the counterfactual inferences more difficult.

---

## [Editor Report · Decision Letter 2]

11 Nov 2020

PONE-D-20-08299R2

How children and adults keep track of real information when thinking counterfactually

PLOS ONE

Dear Dr. Gómez-Sánchez,

Thank you for submitting your manuscript to PLOS ONE. After careful consideration, we feel that it has merit but does not fully meet PLOS ONE’s publication criteria as it currently stands. Therefore, we invite you to submit a revised version of the manuscript that addresses the minor points I list in my comments below.

We look forward to receiving your revised manuscript.

Kind regards,

Emmanuel Manalo, PhD

Academic Editor

PLOS ONE

Additional Editor Comments (if provided):

Thank you for the revisions you have made according to Reviewer 1's comments and suggestions.

I have now carefully checked through your Revision 2 manuscript and found a few more minor corrections that need to be made before we can accept this for publication. I list those here. Please make these corrections, highlight them clearly in the manuscript, and I will quickly check them before making a final decision on your paper.

Line 84: same way **as** adults ...

Line 238: are expected **in** detecting ...

Line 482: **was** to evaluate ...

Line 488: this last aspect **had** not been ...

Line 506: Additionally, we **aimed** to dissociate ...

Lines 575-577: We carried out an ANOVA with Conditional (within-subject variable) and Age (between-subject variable) as the independent variables, and Inferential accuracy as the dependent variable.

Line 648: **In** our research, ...

---

## [Author Response · Author response to Decision Letter 2]

12 Nov 2020

All the corrections below proposed by the Editor have been made:

• Line 84: same way as adults ...

• Line 238: are expected in detecting ...

• Line 482: was to evaluate ...

• Line 488: this last aspect had not been ...

• Line 506: Additionally, we aimed to dissociate ...

• Lines 575-577: We carried out an ANOVA with Conditional (within-subject variable) and Age (between-subject variable) as the independent variables, and Inferential accuracy as the dependent variable.

• Line 648: In our research, ...

---

## [Editor Report · Decision Letter 3]

13 Nov 2020

How children and adults keep track of real information when thinking counterfactually

PONE-D-20-08299R3

Dear Dr. Gómez-Sánchez,

We’re pleased to inform you that your manuscript has been judged scientifically suitable for publication and will be formally accepted for publication once it meets all outstanding technical requirements.

Kind regards,

Emmanuel Manalo, PhD

Academic Editor

PLOS ONE
---

## [Editor Report · Acceptance letter]

25 Nov 2020

PONE-D-20-08299R3 

How children and adults keep track of real information when thinking counterfactually 

Dear Dr. Gómez-Sánchez:

I'm pleased to inform you that your manuscript has been deemed suitable for publication in PLOS ONE. Congratulations! Your manuscript is now with our production department. 

Kind regards, 

on behalf of

Professor Emmanuel Manalo 

Academic Editor

PLOS ONE